# Comparative Analysis of Plastomes in Elsholtzieae: Phylogenetic Relationships and Potential Molecular Markers

**DOI:** 10.3390/ijms242015263

**Published:** 2023-10-17

**Authors:** Xiong-De Tu, Zhuang Zhao, Cheng-Yuan Zhou, Meng-Yao Zeng, Xu-Yong Gao, Ming-He Li, Zhong-Jian Liu, Shi-Pin Chen

**Affiliations:** 1College of Forestry, Fujian Agriculture and Forestry University, Fuzhou 350002, China; 2Key Laboratory of National Forestry and Grassland Administration for Orchid Conservation and Utilization at Landscape Architecture and Arts, Fujian Agriculture and Forestry University, Fuzhou 350002, China

**Keywords:** plastid genome, comparative analysis, Elsholtzieae, *Mosla*, phylogenomics

## Abstract

The Elsholtzieae, comprising ca. 7 genera and 70 species, is a small tribe of Lamiaceae (mint family). Members of Elsholtzieae are of high medicinal, aromatic, culinary, and ornamentals value. Despite the rich diversity and value of Elsholtzieae, few molecular markers or plastomes are available for phylogenetics. In the present study, we employed high-throughput sequencing to assemble two *Mosla* plastomes, *M. dianthera* and *M. scabra*, for the first time, and compared with other plastomes of Elsholtzieae. The plastomes of Elsholtzieae exhibited a quadripartite structure, ranging in size from 148,288 bp to 152,602 bp. Excepting the absence of the pseudogene *rps19* in *Elsholtzia densa*, the exhaustive tally revealed the presence of 132 genes (113 unique genes). Among these, 85 protein-coding genes (CDS), 37 tRNA genes, 8 rRNA genes, and 2 pseudogenes (*rps19* and *ycf1*) were annotated. Comparative analyses showed that the plastomes of these species have minor variations at the gene level. Notably, the *E. eriostchya* plastid genome exhibited increased GC content regions in the LSC and SSC, resulting in an increased overall GC content of the entire plastid genome. The *E. densa* plastid genome displayed modified boundaries due to inverted repeat (IR) contraction. The sequences of CDS and intergenic regions (IGS) with elevated variability were identified as potential molecular markers for taxonomic inquiries within Elsholtzieae. Phylogenetic analysis indicated that four genera formed monophyletic entities, with *Mosla* and *Perilla* forming a sister clade. This clade was, in turn, sister to *Collinsonia*, collectively forming a sister group to *Elsholtzia*. Both CDS, and CDS + IGS could construct a phylogenetic tree with stronger support. These findings facilitate species identification and DNA barcoding investigations in Elsholtzieae and provide a foundation for further exploration and resource utilization within this tribe.

## 1. Introduction

*Mosla* (Benth.) Buch.-Ham. ex Maxim. is an important genus in tribe Elsholtzieae of Lamiaceae, including 14 species, mainly distributed in East and Southeast Asia, while in North America, *M. dianthera* and *M. scabra* were introduced and cultivated [1]. *M. dianthera* and *M. scabra* are traditional Chinese medicines [2,3,4], and studies on the active ingredients in their extracts have also confirmed their effective treatment of diseases [3,5,6,7]. The tribe Elsholtzieae encompasses seven distinct genera: *Collinsonia* L., *Elsholtzia* Willd., *Keiskea* Miq., *Mosla* (Benth.) Buch.-Ham. ex Maxim., *Ombrocharis* Hand.-Mazz., *Perilla* L., and *Perillula* Maxim. These genera are widely distributed in East and Southeast Asia, but *Collinsonia* has four species distributed in eastern North America [8]. This tribe includes more than 70 species, some of which have high medicinal, aromatic, and culinary value. Species such as *P. frutescens* not only have medicinal value but are also used as a food spices and garden ornamental plants [4,9,10].

Pollen morphology studies of representative species of *Mosla* and closely related genera, namely, *Collinsonia*, *Elsholtzia*, *Keiskea*, *Perilla*, and *Perillula*, showed that *Mosla* (except *M. cavaleriei*) and *Perilla* are the most closely related in these genera [11]. Chen et al. (2016) sampled from all seven genera, employing four chloroplast fragments (*trnL-F*, *ycf1*, *rps15-ycf1*, and *rpl32-trnL*) and two nuclear fragments (ITS and ETS) to construct the Elsholtzieae phylogeny [12]. Their outcomes demonstrated that phylogeny constructed by the cpDNA offered higher support for Elsholtzieae to form a separate clade and a clearer inter-generic relationship than the nrDNA approach. In the phylogenetic tree constructed by cpDNA, the results showed *Mosla* to be a sister of the *Perilla* + *Keiskea* clade, and subsequently, these three genera were sistered to *Collinsonia* with high support. Li et al. (2017) is the most comprehensive study on intra-tribe sampling to date, using a combination of two nuclear (ITS and ETS) fragments and five chloroplast (*rbcL*, *matK*, *trnL-F*, *ycf1*, and *rps15-ycf1*) fragments to reconstruct the phylogeny of Elsholtzieae [13]. The results showed that Elsholtzieae consisted of five clades, and *Collinsonia* was established as a monophyletic entity and a sister to the clade encompassing *Mosla*, *Keiskea*, and *Perilla*. Intriguingly, *Perilla* was found embedded within *Keiskea.* Obviously, the phylogeny of Elsholtzieae constructed using the molecular data of a small number of markers is not robust, and it is not consistent with the previous view that *Perilla* is closest to *Mosla* based on palynological characteristics.

In recent years, propelled by the advancement of high-throughput sequencing technology, chloroplast genomes, also referred to as plastid genomes, have gained escalating prominence in delineating the phylogenetics of the Lamiaceae [14,15,16,17]. Zhao et al. (2021) used 79 plastome-coding regions to construct phylogenetic relationships on large-scale sampling data of Lamiaceae [14]. Their findings underscored the robust support that plastome-coding regions confer both deep and shallow nodes in the phylogenetic analysis. Notably, their examination encompassed three Elsholtzieae genera (*Perilla*, *Collinsonia*, and *Elsholtzia*). This inquiry demonstrated the monophyly of Elsholtzieae, wherein *Perilla* and *Collinsonia* are sisters to the *Elsholtzia*. Wu et al. (2021) used the plastomes to study the phylogeny within the genus *Salvia* to provide evidence that the plastomes can resolve the phylogenetic relationship within the genus of Lamiaceae [15]. For the phylogeny of Elsholtzieae, more sampled plastomes are necessary in order to resolve the inter-generic relationships within Elsholtzieae. However, only two genera (*Elsholtzia* and *Perilla*) of Elsholtzieae are publicly available (NCBI GenBank database, accessed on 1 August 2023). Therefore, the existing research on the characteristics of the plastids of the tribe Elsholtzieae has not been developed, and the field of reconstructing the phylogenetic tree of the Elsholtzieae from the plastid genome is still very sparse.

The Lamiaceae, also known as the mint family, is the sixth largest family of angiosperms and has long been known for its aromatic oils. Elsholtzieae is a minor member of the Lamiaceae. For a comprehensive understanding of the evolution of plastomes in the Elsholtzieae, 18 plastomes of Elsholtzieae were used in this study. By scrutinizing plastomes disparities, pinpointing mutational hotspots, dissecting codon compositions, demarcating repeat regions, and identifying high variable regions within coding and intergenic regions, we elucidated the level of the plastid genome evolution model. Moreover, our analysis revealed the phylogenetic relationships of Elsholtzieae through the plastid genome. Therefore, our study provides valuable information for plastid genome evolution, phylogenetic relationship, and species identification of Elsholtzieae.

## 2. Results

### 2.1. Plastome Features

The complete plastome sequences of *M. dianthera* and *M. scabra* exhibit the typical quadripartite structure comprising two inverted repeats (IRs), a large single-copy (LSC) region, and a small single-copy (SSC) region (Figure 1). The plastid genomes lengths of *M. dianthera* and *M. scabra* were 152,550 bp and 152,562 bp, respectively. By comparing the published plastomes of Elsholtzieae, we observed that the plastid length of *Mosla* is similar to that of *Perilla* and *Collinsonia*, which is 152 kb, while the length of the *Elsholtzia* ranged from 148 kb to 151 kb (Appendix A). *E. eriostachya* has the smallest plastid length at 148,288 bp, whereas *P. frutescens* var. *hirtella* boasts the longest length among Elsholtzieae at 152,602 bp. A significant correlation was noted between the LSC region and the complete plastid genome length, as well as between IR and the complete plastid genome length in Elsholtzieae (Appendix A). Except *E. eriostachya*, the total GC content is 37.75–37.96%, in which the GC content of the IR regions (43.08–43.18%) is visibly greater than that of the LSC (35.74–36.03%) and SSC (31.71–32.00%) regions (Appendix A). It is noteworthy that the GC content of the complete plastid genome, LSC, and SSC of *E. eriostachya* is the highest, being 38.18%, 36.27%, and 32.37%, respectively.

With the exception of the *rps19* pseudogene deletion in *E. densa*, 132 genes (113 unique genes) were detected, including 85 protein-coding genes, 37 tRNA genes, 8 rRNA genes, and 2 pseudogenes (*rps19*, and *ycf1*) (Table 1). A total of 18 unique genes containing introns were observed in the studied species, of which 15 contained 1 intron (*atpF*, *petB*, *petD*, *ndhA*, *ndhB*, *trnA-UGC*, *trnG-UCC*, *trnI-GAU*, *trnK-UUU*, *trnL-UAA*, *trnV-UAC*, *rpoC1*, *rps16*, *rpl2*, *rpl16*), whereas 3 unique genes (*rps12*, *clpP* and *ycf3*) contained 2 introns (Table 1). No rearrangements in gene organization were found in the analyzed plastomes (Appendix A).

### 2.2. Expansion and Contraction of IRs

Comparative sequence analysis of Elsholtzieae indicated that with the exception of *E. densa*, the SC/IR boundary region is conservative (Figure 2, Appendix A). In *E. densa*, the *rps19* pseudogene is entirely situated in the LSC region, with a 9 bp separation from the LSC/IRb (JLB) boundary. There is no duplication of the *rps19* gene in IRa, indicating that the IR region of *E. densa* has contracted. However, in other species of Elsholtzieae, the *rps19* gene is located within the JLB boundary, with 43–68 bp spanning into the IRb region (Figure 2, Appendix A). Correspondingly, the *rps19* gene is partially present in IRa; thus, a pseudogene of *rps19* exists. We also found that the protein-coding sequence of the gene *ycf1* spans the SSC/IRa (JSA) boundary, generating a pseudogenic version of *ycf1* at IRb.

### 2.3. Codon Usage Analysis

Based on the sequences of 79 protein-coding genes from 9 representative plastomes of Elsholtzieae, the codon usage frequency and relative synonymous codon usage (RSCU) were calculated (Appendix A). The total number of codons for protein-coding genes in the plastomes ranged from 22,594 in *E. eriostachya* to 22,953 in *E. rugulosa* (Appendix A) among the nine plastomes of Elsholtzieae. Further codon analysis showed that the nine plastomes have similar codon constituents and close RSCU values. We identified a total of 61 synonymous codons, excepting stop codons, among which we identified a total of 30 codons with RSCU > 1 and 34 codons with RSCU < 1 (Appendix A). The majority of amino acid codons exhibited a bias, although codons AU(T)G and U(T)GG, which encode methionine (Met) and tryptophan (Trp), respectively, both showed no codon preferences (RSCU = 1.00). Leucine (Leu: 10.61–10.72%) was the most commonly encoded amino acid in all plastomes, while cysteine (Cys: 1.11–1.13%) was the least abundant (Appendix A). In protein-coding genes of the plastomes, 69.97–70.62% of all codons were terminated with A/U, indicating a bias for A/U(T) bases (Appendix A).

### 2.4. Repeat Sequence Analysis

A total of 422 SSRs were detected in the plastomes of the nine representative plastomes of Elsholtzieae, ranging from 38 (*E. eriostachya*) to 60 (*E. angustifolia*) per plastome (Figure 3). *M. dianthera* and *M. scabra* had the same number of SSRs, while the number of SSRs between *P. frutescens* and its cultivars was similar. However, the number of SSRs among *Elsholtzia* species varied considerably, representing the range of SSRs numbers within Elsholtzieae. Four SSR types (mono-, di-, tri-, and tetra- repeats) all appeared in Elsholtzieae species. Mononucleotide repeats are most abundant (67.78% of the total SSRs), followed by dinucleotide repeats (14.69%), whereas pentanucleotides repeats are very rare among these plastomes, only found in *E. angustifolia* and *E. rugulosa*. For all plastomes analyzed, SSRs are located mainly in the LSC and IGS (Appendix A).

Four categories of repeats (forward, palindromic, complement, and reverse repeats) were identified in the nine plastomes of Elsholtzieae, totaling 401 repeats (Appendix A). Forward repeats (50.37%) and palindromic repeats (46.13%) are the most abundant in the total repetitions and exist in all plastids, while complement repeats (1.50%), and reverse repeats (2.00%) are less abundant in the total repetitions and only exist in some plastids (Appendix A). The lowest number of repeats was observed in *E. eriostachya* (27), and the highest number occurred in *E. rugulosa* (56). We artificially divided all repeats into four categories (30–39 bp, 40–49 bp, 50–59 bp, and >60 bp) based on their length. Of these, 288 (71.82%) have lengths of 30–39 bp, followed by 92 (22.94%) with lengths of 40–49 bp, whereas only 9 (2.24%) have lengths of 50–59 bp, and 12 (2.99%) have lengths longer than 60 bp.

### 2.5. Sequence Divergence Analysis

To identify the diversity in the nine representative plastomes of Elsholtzieae, mVISTA was used to align plastome sequences with *E. eriostachya* as a reference (Figure 4). Alignment revealed that protein-coding genes were more conserved than conserved non-coding sequences, and the two IR regions were less divergent than the LSC and SSC regions. For a deeper understanding of the sequence divergence among these 18 plastomes of Elsholtzieae, both coding regions and intergenic regions were extracted to calculate nucleotide variability (Pi) (Figure 5; Appendix A). The Pi values for intergenic regions were approximately twice as high as or higher than those for coding regions (Figure 5). In comparison to most fragments in the LSC and SSC regions, the Pi values of IR regions are much lower in both the coding and intergenic regions. In the coding regions of the Elsholtzieae, eight mutational hotspot regions (*matK*, *psbT*, *rpl22*, *ndhF*, *rpl32*, *ccsA*, *ndhD*, and *ycf1*) were observed (Figure 5A). For intergenic regions, five mutational hotspot regions (*trnH-GUG_psbA*, *trnS-GCU_trnG-UCC*, *trnS-GGA_rps4*, *ccsA_ndhD*, and *ndhG_ndhI*) were identified (Figure 5B).

### 2.6. Phylogenetic Analyses

To elucidate the phylogenetic relationship of Elsholtzieae, four matrices, complete plastid genomes, protein-coding genes (CDS), intergenic regions (IGS), and CDS + IGS of 18 samples in Elsholtzieae and two outgroups were used to construct the phylogenetic trees (Figure 6). Both the maximum likelihood (ML) and Bayesian inference (BI) phylogenetic trees, based on the four matrices, exhibited the same topological structure, showing that the four genera are monophyletic. *Mosla* and *Perilla* form a clade and then become sisters with *Collinsonia*, and these three genera function as sisters to *Elsholtzia*. Among the four tree-building strategies, the support rate of the *E. densa* + *E. eriostachya* clade to the remaining eight *Elsholtzia* is low, the difference is large, and the support rates of the other nodes are all high (Figure 6). The highest support for this node was observed in the phylogenetic tree constructed using CDS + IGS (94/91/1.00) (Figure 6C), followed by the phylogenetic tree constructed using CDS (90/86/1.00) (Figure 6A), and the lowest phylogenetic tree constructed for complete plastid genomes (10/52/0.73) is shown in Figure 6D.

## 3. Discussion

### 3.1. Variation of Plastome Sequences

This study assembled and annotated two *Mosla* plastomes for the first time, which are similar to most angiosperms, featuring a typical quadripartite structure [18]. The length of the plastid genome of *Mosla* is comparable to that of *Perilla* and *Collinsonia* because these three genera are more closely related. We found a significant correlation between the LSC region and the length of the complete plastid genome, and between the IR and the length of the complete plastid genome in Elsholtzieae, which is consistent with other studies [19]. In angiosperms, plastid gene loss and gain events and structural rearrangements have been identified in some species or genera [20], but these events have not been observed in Elsholtzieae. These results suggest that the plastid structure and gene number of Elsholtzieae are largely conserved.

The GC content plays a pivotal role in the evolution of genomic structures [21]. With the exception of *E. eriostachya*, the GC content (LSC, SSC, and IR) was similar across species. The plastid genome of *E. eriostachya*, as the shortest plastid genome of Elsholtzieae, has the highest GC content, indicating that the low GC content regions of LSC and SSC of *E. eriostachya* were lost, resulting in an increase in the overall GC content. We also found that the IR region had significantly higher GC content than other regions, mainly due to the high GC content of four rRNA genes (*rrn23*, *rrn16*, *rrn5*, and *rrn4.5*) [22].

The expansion and contraction of the IR boundary is the main cause of the change in the size of the plastid genome and plays an important role in the evolution of species [23]. In this study, we observed the relationship between the LSC, SSC, and IR regions in the 18 plastomes of Elsholtzieae, except that IR contraction occurred in *E. densa*, resulting in changes in IR boundaries, and most of the others were found to have consistent IR boundaries, including its variant *E. densa* var. *ianthina*. Previous reports suggested that the different lengths of the plastid genome may result from the expansion and contraction that occurs in the IR region [24,25]. In this study, the change in the SC/IR boundary of *E. densa* may be caused by IR contraction.

### 3.2. Codon Usage and Repeat Sequence Analysis

Codon usage bias plays a significant role in the evolution of plastomes and affects the expression of gene functions [26]. Organisms with close genetic relationships have very similar codon use biases [27]; through comparative analysis of RSCU values, codon usage bias in the plastid genome of Elsholtzieae is very similar. Codon usage in plastomes is usually biased toward codons ending in A or T(U) [28]. This bias was also observed in the plastomes of Elsholtzieae. According to the RSCU analysis, it was found that most of the frequently used codons (RSCU > 1) were A/U-ending, whereas the less frequently used codons (RSCU <  1) were C/G-ending. This was consistent with the results of the base composition analysis.

The SSRs present in the plant plastid genome can be used to study the genetic diversity of plant populations, understand the evolutionary relationship between plant species, reconstruct their phylogenetic history, identify SSR markers associated with desired traits for plant breeding, and develop improved crop varieties [29,30,31]. Analysis of the nine representative plastomes of Elsholtzieae revealed five types of SSRs (mono-, di-, tri-, tetra-, and penta- repeats). Among them, the A/T single-nucleotide repeat sequence was the most abundant, while the G/C single-nucleotide repeat sequence was least abundant in the complete plastid genome. This phenomenon is consistent with many angiosperm plastids and may be the result of plastid bias towards A/T [29,32]. Large duplications are crucial for studying genome recombination, rearrangement, and phylogeny, and for causing substitutions and insertions in plastid genomes [33]. Among the 401 repeats identified in this study, forward repeats and palindromic repeats between 30 and 39 bp were the most common, consistent with many plastid studies [19,34]. These results have a guiding significance in the identification and analysis of genetic diversity in Elsholtzieae plants.

### 3.3. Barcoding Investigation

Determining variation in plastid genomes is critical for understanding the structure and evolution of plastid genomes. In this study, we observed lower variability in the IR region compared to the LSC and SSC regions, consistent with findings from previous studies in Lamiaceae and other angiosperms. We identified highly variable regions in the coding region and the intergenic region, respectively. Since the mutation rate in the non-coding region is generally higher than that in the coding region [35], the high variation Pi in the intergenic region of the plastid genome of Elsholtzieae is about two times higher than that in the coding region. Through the calculation and comparison of Pi values, we identified several hotspot regions, including *matK*, *ycf1*, *trnH-GUG*_*psbA*, *trnS-GCU*_*trnG-UCC*, *trnS-GGA*_*rps4*, *ccsA*_*ndhD*, and *ndhG*_*ndhI*. However, it is worth noting that in previous phylogenetic studies [12,13,36], only three of these plastomes’ DNA sequences (*matK*, *ycf1*, and *trnH-GUG*_*psbA*) were employed. These regions have significant variability and can be used as valuable molecular markers in future phylogenetic studies of Elsholtzieae to help identify Elsholtzieae plants.

### 3.4. Phylogenetic Analysis

The phylogenetic relationship of the Elsholtzieae family is mainly based on a small number of molecular markers. Peirson (2003) studied the phylogeny of Elsholtzieae using nrITS, and the results showed that *Collinsonia* is sister to the clade formed by *Keiskea* and *Perilla*, and *Elsholtzia* is the sister of these three genera [37]. Chen et al. (2016) and Li et al. (2017) have made some progress on the phylogeny of Elsholtzieae [12,13]. However, their results show that there are conflicts in the phylogenetic tree relationship; the topology is unstable, and the inter-generic and intra-generic relationships are unclear. The significance of plastid genomes in reconstructing phylogenetic relationships and comprehending evolutionary history has been well-established and is increasingly applied in Lamiaceae research [14,15,16,17]. Zhao et al. (2021) showed that phylogenetic analyses of the Lamiaceae based on plastid coding regions offered robust support with respect to both deep and shallow nodes [14]. In order to explore the phylogeny location of *Mosla* and clearly elucidate the genetic evolutionary relationships within Elsholtzieae, we performed phylogenetic analysis based on the plastomes of *M. dianthera* and *M. scabra* and 16 other plastomes. We constructed the Elsholtzieae phylogenetic tree through the four matrices of the plastomes (complete plastid genome, CDS, IGS, and CDS + IGS), and the results highly support that the *Mosla* + *Perilla* clade is sister to *Collinsonia*, and that these three genera are sisters to *Elsholtzia*. The clade comprising *E. densa* + *E. eriostachya* as sisters to the remaining Elsholtzia plants exhibits varying levels of support across the four tree-building strategies, with the highest support observed in the phylogenetic tree constructed using CDS + IGS (94/91/1.00), followed by CDS (90/86/1.00). Moderate support for this node is reflected in the phylogenetic tree constructed using a small number of molecular markers [12,13]. The combination of the CDS + IGS matrix can provide a reference for the construction of the phylogenetic relationship of this tribe and solve the disputes, such as whether to create a new genus *Vuhuangia* Solomon Raju, Molinari, and Mayta independently from *Elsholtzia* with *E. flava* + *E. penduliflora*, and whether *Keiskea* is monophyletic.

## 4. Materials and Methods

### 4.1. Plant Materials, DNA Extraction and Sequencing

Healthy and fresh leaves of *M. dianthera* and *M. scabra* were collected from mature plants in Mingxi district, Fujian, China, and deposited as voucher specimens at the Herbarium of Fujian Agriculture and Forestry University in Fuzhou, China. Total genomic DNA was extracted from fresh leaves using a modified CTAB method as described by Doyle [38]. The purified DNA samples were sheared into fragments with an average length of 350 bp for library preparation following the manufacturer’s guidelines (Illumina, San Diego, CA, USA). Following a quality inspection of the library, paired-end (PE) sequencing with 150 bp read length was performed using the Illumina HiSeq-2500 platform at the Beijing Genomics Institute (Shenzhen, China).

### 4.2. Plastid Genome Assembly and Annotation

The quality of the raw PE reads was verified by the Trimmomatic v.0.32 [39] with default settings to obtain high-quality clean reads. Paired-end reads from the clean data were assembled into contigs using GetOrganelle v1.7.1 [40], and the plastomes were annotated by GeSeq [41], manually adjusted by Geneious 11.1.5 [42]. To avoid confusion of annotation, we re-annotated the 16 plastomes obtained from GenBank. The newly assembled plastomes obtained in this study underwent thorough examination and validation before submission to GenBank, ensuring its accuracy and error-free status. Circular visualization of plastomes was carried out using the online tool OGDRAW v.1.3.1 (https://chlorobox.mpimp-golm.mpg.de/OGDraw.html (accessed on 21 April 2023)) [43].

### 4.3. Genome Structure Comparisons and Sequence Divergence Analysis

Rearrangement analysis of the 18 plastomes in Elsholtzieae was conducted using progressive Mauve v.2.4.0 [44]. Wholegenome alignment of the nine representative plastomes of Elsholtzieae was performed and plotted with the mVISTA program (http://genome.lbl.gov/vista/mvista/submit.shtml (accessed on 27 April 2023)) [45] in the Shuffle-LAGAN model, with *E. rugulosa* as the reference. The IR/SC boundaries of these plastomes were compared using the Perl script CPJSdraw.pl (https://github.com/xul962464/CPJSdraw (accessed on 29 June 2023)). The nucleotide diversity (Pi) of the plastomes was evaluated using DnaSP v.6.12 software [46].

### 4.4. Repeat Sequence and Codon Usage Analysis

The length and location of forward, palindromic, reverse, and complement repeats in the plastomes of Elsholtzieae were detected by the REPuter program (https://bibiserv.cebitec.uni-bielefeld.de/reputer (accessed on 26 April 2023)) [47]. These four types of repeats were detected with a minimum repeat size of 30 bp, an edit distance of 3, and 90% similarities. The simple sequence repeats (SSR) were determined by the search tool MISA (https://webblast.ipk-gatersleben.de/misa/ (accessed on 28 April 2023)) [48] with thresholds of 10, 5, 4, 3, 3, and 3 for mononucleotides, dinucleotides, trinucleotides, tetranucleotides, pentanucleotides, and hexanucleotides, respectively. Composite microsatellites were identified by setting the minimum distance between the two SSRs to be less than 100 bp. Codon usage and relative synonymous codon usage (RSCU) values were calculated using Codon W (http://codonw.sourceforge.net/ (accessed on 28 June 2023)) [49]. Repeat sequences of protein-coding regions were eliminated from the codon usage calculations to avoid sampling errors.

### 4.5. Phylogenetic Analyses

To construct the phylogeny of Elsholtzieae, we used the 18 plastomes of Elsholtzieae, including 1 *Collinsonia*, 11 *Elsholtzia*, 2 *Mosla*, and 4 *Perilla*. Two species of the Ocimeae, *Hanceola exserta* and *Siphocranion macranthum* were set as the outgroups (Appendix A). CDS and IGS were extracted from the 20 plastomes described above using PhyloSuite v.1.1.16 [50]. By removing repetitive sequences, the final 79 CDS regions and 66 IGS regions were used to construct the phylogenetic tree. The sequences were aligned by MAFFT v7.490 [51] with auto parameters. We used the complete plastid genome, 79 CDS, 66 IGS, and 79 CDS + 66 IGS matrices to construct the Elsholtzieae phylogeny, respectively. ML analyses were performed in IQ-TREE v2.0.3 [52], incorporating the SH-aLRT test and ultrafast bootstrap (UFBoot) feature, using the model identified by ModelFinder implemented in IQ-TREE (-alrt 1000 -bb 1000 -m MFP). BI analysis was performed with MrBayes v3.2.7 [53] using best-fit models (GTR + I + G), which were selected with MrModeltest 2.4 [54] using AIC. The following settings were used: ngen = 2,000,000; samplefreq = 1000; burninfrac = 0.25.

## 5. Conclusions

This study assembled and annotated two *Mosla* plastomes for the first time and compared them with those of other Elsholtzieae plants to investigate plastid genome differences within the tribe. The plastomes of Elsholtzieae exhibited a typical quadripartite structure, comprising an LSC region, an SSC region, and a pair of IRs. Plastomes lengths ranged from 148,288 bp (*E. eriostachya*) to 152,602 bp (*P. frutescens* var. *hirtella*). In addition to the *rps19* pseudogene, which was absent in *E. densa* due to IR contraction, we detected 132 genes (113 unique genes), including 85 protein-coding genes, 37 tRNA genes, 8 rRNA genes, and 2 pseudogenes (*rps19*, and *ycf1*). Several highly variable regions were identified in the CDS and IGS of the Elsholtzieae plastid genome, such as *matK*, *ycf1*, *trnH-GUG*_*psbA*, *trnS-GCU*_*trnG-UCC*, *trnS-GGA*_*rps4*, *ccsA*_*ndhD*, and *ndhG*_*ndhI*. These regions could potentially serve as markers for phylogenetic analysis. Our phylogenetic analysis indicated that matrices constructed using CDS + IGS provided stronger support. The results corroborate the *Mosla* + *Perilla* clade as sisters to *Collinsonia*, with these three genera forming a sister group to *Elsholtzia*. This study presents novel evidence supporting the utility of plastid genomes in elucidating the phylogeny of Elsholtzieae, particularly with expanded species sampling, aiding in the resolution of classification challenges within this tribe.

## Figures and Tables

**Figure 1 ijms-24-15263-f001:**
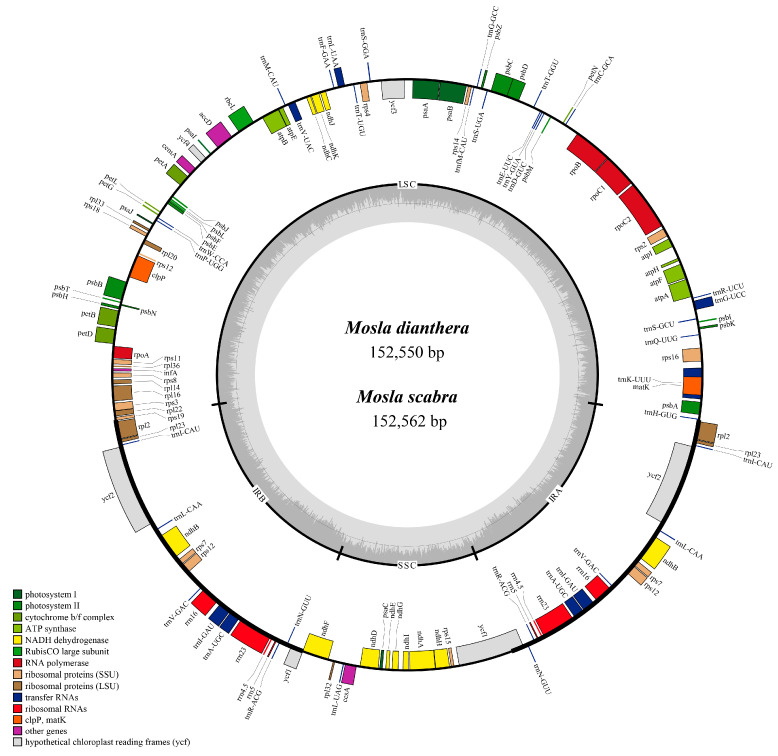
The annotation map of *M. dianthera* and *M. scabra* plastomes. The darker gray in the inner circle corresponds to the GC content. The inner circle presents different shades of gray, with lighter shades representing AT content and darker shades representing GC content of the plastome.

**Figure 2 ijms-24-15263-f002:**
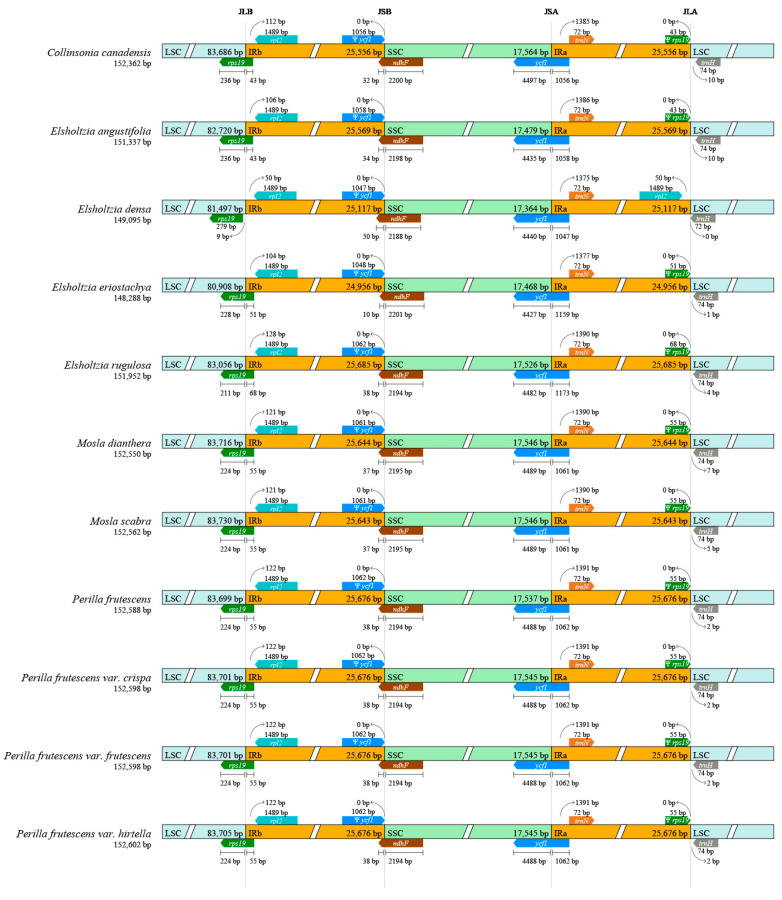
Comparison of junctions between the LSC, SSC, and IRs in the 11 Elsholtzieae plastomes. Distance in this figure is not to scale. Pseudogenes are marked by Ψ. JLB, JSB, JSA, and JLA denoted the junction sites of LSC/IRb, IRb/SSC, SSC/IRa, and IRa/LSC, respectively.

**Figure 3 ijms-24-15263-f003:**
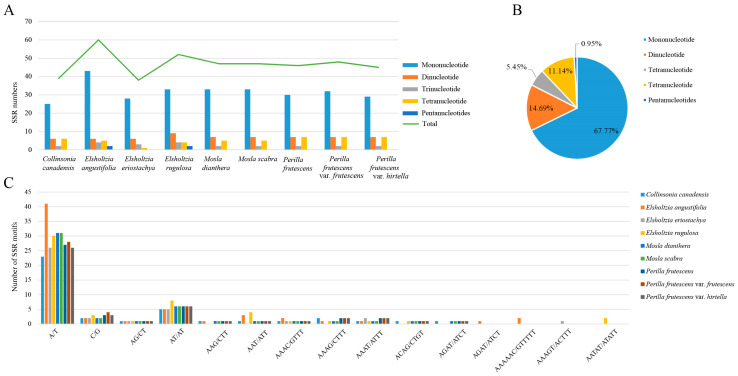
Analyses of simple sequence repeats (SSR) in the nine Elsholtzieae plastomes. (**A**) number of SSRs and their types; (**B**) percentage of SSR types; (**C**) number of SSR motifs.

**Figure 4 ijms-24-15263-f004:**
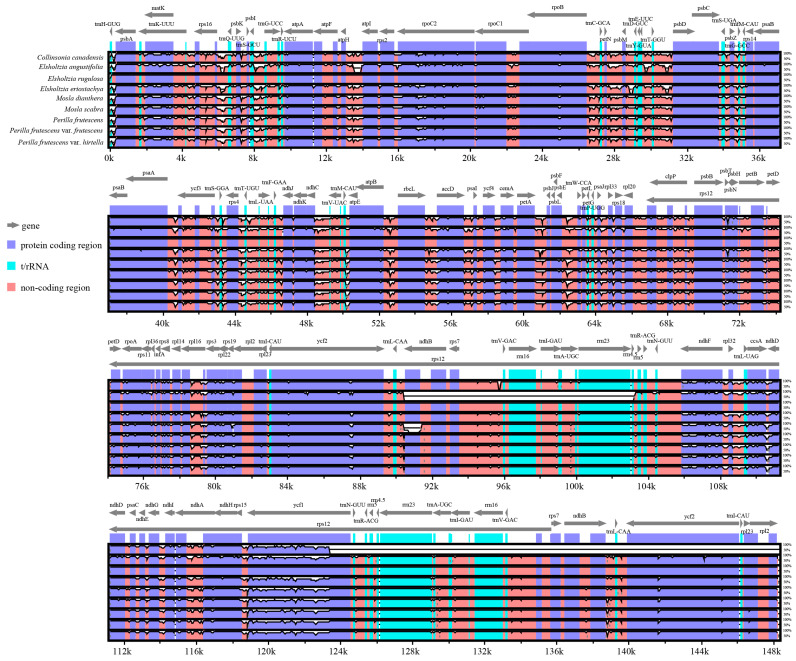
Sequence alignment of nine Elsholtzieae plastomes using the mVISTA program with *E. rugulosa* as a reference. The X- and Y-scales represent the coordinates within plastomes and the percentage of identity (50~100%), respectively. Gray arrows indicate the direction of transcription of each gene. Genome regions are color-coded as protein-coding region, tRNA, rRNA, and conserved non-coding region.

**Figure 5 ijms-24-15263-f005:**
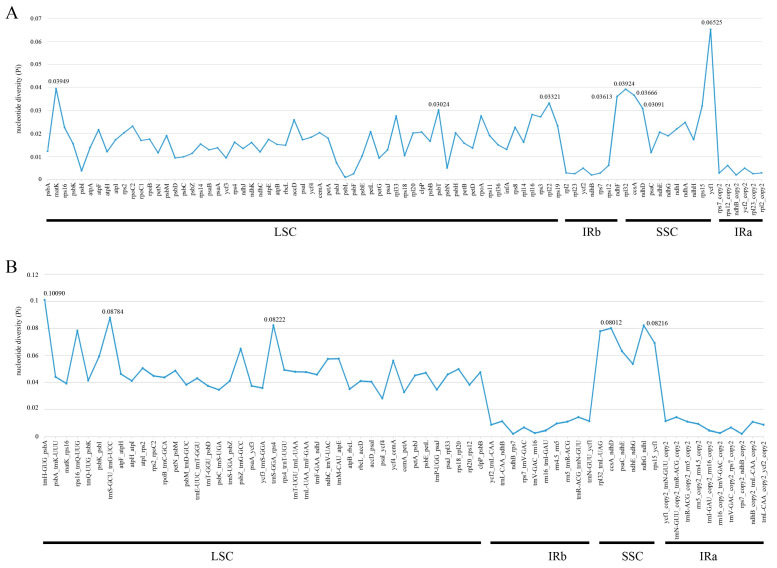
The nucleotide diversity (Pi) values of the aligned Elsholtzieae plastomes. (**A**) protein-coding genes; (**B**) intergenic regions.

**Figure 6 ijms-24-15263-f006:**
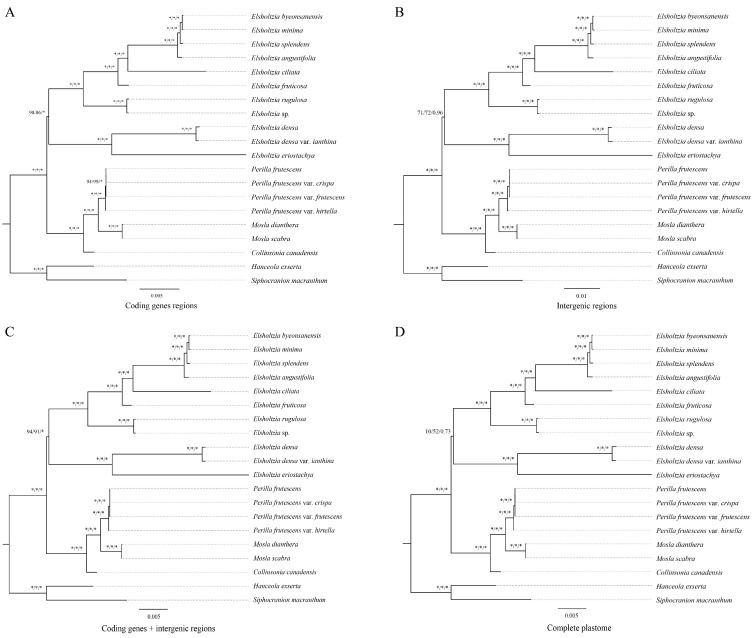
Phylogenetic tree of Elsholtzieae based on plastome. Numbers near the nodes are bootstrap percentages and Bayesian posterior probabilities (SH-aLRT left, UFBoot middle, and PP right). An asterisk (*) indicates that the node is 100% bootstrap or 1.00 posterior probability. (**A**) Phylogram based on coding genes regions (CDS); (**B**) phylogram based on intergenic regions (IGS); (**C**) phylogram based on coding genes + intergenic regions (CDS + IGS); (**D**) phylogram based on complete plastomes.

**Table 1 ijms-24-15263-t001:** Genetic classification of the plastomes of Elsholtzieae.

Category of Genes	Group of Genes	Gene Name
Photosynthesis-related genes	Photosystem I	*psaA*, *psaB*, *psaC*, *psaI*, *psaJ*
	Photosystem II	*psbA*, *psbB*, *psbC*, *psbD*, *psbE*, *psbF*, *psbH*, *psbI*, *psbJ*, *psbK*, *psbL*, *psbI*, *psbM*, *psbN*, *psbT*, *psbZ*
	Large subunit of rubisco	*rbcL*
	Cytochrome b/f complex	*petA*, *petB **, *petD **, *petG*, *petL*, *petN*
	ATP synthase	*atpA*, *atpB*, *atpE*, *atpF **, *atpH, atpI*
	NADH dehydrogenase	*ndhA **, *ndhB ** (*2), *ndhC*, *ndhD*, *ndhE*, *ndhF*, *ndhG*, *ndhH*, *ndhI*, *ndhJ*, *ndhK*
Self-replication-related genes	Ribosomal RNA	*rrn4*.*5* (*2), *rrn5* (*2), *rrn16* (*2), *rrn23* (*2)
	Transfer RNA	*trnA-UGC ** (*2), *trnC-GCA*, *trnD-GUC*, *trnE-UUC*, *trnF-GAA*, *trnfM-CAU*, *trnG-GAA*, *trnG-UCC **, *trnH-GUG*, *trnI-CAU* (*2), *trnI-GAU ** (*2), *trnK-UUU **, *trnL-CAA* (*2), *trnL-UAA **, *trnL-UAG*, *trnM-CAU*, *trnN-GUU* (*2), *trnP-UGG*, *trnQ-UUG*, *trnR-ACG* (*2), *trnR-UCU*, *trnS-GCU*, *trnS-GGA*, *trnS-UGA*, *trnT-GGU*, *trnT-UGU*, *trnV-GAC* (*2), *trnV-UAC* *, *trnW-CCA*, *trnY-GUA*
	RNA polymerase	*rpoA*, *rpoB*, *rpoC1 **, *rpoC2*
	Small subunit of ribosomal proteins	*rps2*, *rps3*, *rps4*, *rps7* (*2), *rps8*, *rps11*, *rps12 *** (*2), *rps14*, *rps15*, *rps16 **, *rps18*, *rps19* (*2),
	Large subunit of ribosomal proteins	*rpl2 ** (*2), *rpl14*, *rpl16 **, *rpl20*, *rpl22*, *rpl23* (*2), *rpl32*, *rpl33*, *rpl36*
Other genes	Protease	*clpP* **
	Maturase	*matK*
	Envelop membrane protein	*cemA*
	Acetyl-CoA-carboxylase	*accD*
	Translation initiation factor	*infA*
	C-type cytochrome synthesis	*ccsA*
Genes with unknown function	Hypothetical chloroplast reading frames	*ycf1* (*2), *ycf*2 (*2), *ycf3* **, *ycf4*

Genes marked with the * or ** sign are the genes with single or double introns, respectively. The duplicated genes located in IR regions were marked as (*2).

## Data Availability

The two plastome sequences are deposited in GenBank at the NCBI repository, accession numbers OR666444 and OR666444.

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
