# Peer review of "Comparative Analysis of Plastomes in Elsholtzieae: Phylogenetic Relationships and Potential Molecular Markers"

_ijms, 2023, doi:10.3390/ijms242015263_

Round 1
Reviewer 1 Report
In the manuscript, authors sequence and characterize plastomes of two Mosla species (M. dianthera and M. scabra) and then compare plastomes of these Mosla species plus three other genera of Elsholtzieae to obtain robust phylogenetic relationships and potential molecular markers for species within Elsholtzieae tribe.
The work is almost exclusively bioinformatic and very descriptive. It uses innovative high-throughput sequencing data, but I don’ t find it very original. Nevertheless I think it could be useful for DNA species identifications and barcoding within this tribe. Overall, the work is clear and easy to read, results are well explained, discussion is well focused even if sometimes little bit repetitive.
More comments and suggestions are given in the file attached.

Author Response
Thank you very much for taking the time to review this manuscript. Those comments are valuable and very helpful. We have read through comments carefully and have made corrections. Based on the instructions provided in your letter, we uploaded the file of the revised manuscript. According to your opinion, there are generally four:
Comment 1: Sentence grammar, incorrect wording, etc.
Response 1: Thank you for your suggestions. We have corrected the problems of sentence grammar errors, word errors and so on. Please see the attachment.
Comment 2: There is a repetition between the Introduction and the discussion section.
Response 2: Thank you for your suggestions. We have removed and simplified where there is duplication to minimize this duplication, please see the attachment.
Comment 3: The picture is not clear enough.
Response 3: Thank you for your suggestions. We have replaced the image with a higher resolution, please see the attachment.
Comment 4: Are there always 3 asterisks in figure 6? page 9, rows 224 – 229
Response 4: Thank you for your suggestions. Yes, as you can see, with the exception of a few nodes, most of the support is the highest, denoted by 3 asterisks. Please see the attachment.

Reviewer 2 Report
The manuscript is well written and surves up to date data on the plastomes in Elsholtzieae taxon. Apart from missing explanations of abbreviated forms, which shall be explained when appear for the first time, starting from the abstract, no further changes required.
This manuscript is searching for convenient molecular tools for identifying species in the Elsholtzieae tribe. Plastome sequences are a good material for studying phylogenetic differences between various plant taxa, so the topic covered in the manuscript is up-to-date and the results are reliably presented. The next step should be to compare the results between the remaining two tribes within the subfamily Nepetoideae. The conclusions include information about the new discovery of two plastomes for the Mosla species and about proposed sequence regions that can serve as potential markers for polygenetic analyses. The references are appropriate and up-to-date. All figures and tables are well presented. All necessary data are included. I recommend the article for publication.
Author Response
Thank you very much for taking time out of your busy schedule to review this manuscript, and thank you very much for your recognition of our manuscript. According to your opinion, there is mainly one point:
Comment 1: The abbreviation form that appears for the first time lacks explanation, and begins with the explanation of the abstract.
Response 1: Thank you for your suggestions. Upon closer inspection, we found that the abbreviations that appeared for the first time had instructions. We would appreciate more detailed suggestions for changes.
Thank you for your careful review. We wish good health to you, your family, and your community.